# Impact of Non-Coding RNAs on Chemotherapeutic Resistance in Oral Cancer

**DOI:** 10.3390/biom12020284

**Published:** 2022-02-09

**Authors:** Karen Yamaguchi, Tomofumi Yamamoto, Junichiro Chikuda, Tatsuo Shirota, Yusuke Yamamoto

**Affiliations:** 1Laboratory of Integrative Oncology, National Cancer Center Research Institute, 5-1-1 Tsukiji, Chuo-ku, Tokyo 104-0045, Japan; karyamag@ncc.go.jp (K.Y.); toyamamo@ncc.go.jp (T.Y.); 2Department of Oral and Maxillofacial Surgery, Showa University School of Dentistry, Tokyo 145-8515, Japan; tshirota@dent.showa-u.ac.jp; 3Department of Molecular and Cellular Medicine, Institute of Medical Science, Tokyo Medical University, Tokyo 160-0023, Japan; 4Department of Oral Oncology Surgery, Showa University School of Dentistry, Tokyo 145-8515, Japan; chang0309@dent.showa-u.ac.jp

**Keywords:** oral cancer, miRNA, lncRNA, EMT, extracellular vesicles

## Abstract

Drug resistance in oral cancer is one of the major problems in oral cancer therapy because therapeutic failure directly results in tumor recurrence and eventually in metastasis. Accumulating evidence has demonstrated the involvement of non-coding RNAs (ncRNAs), such as microRNAs (miRNAs) and long non-coding RNAs (lncRNAs), in processes related to the development of drug resistance. A number of studies have shown that ncRNAs modulate gene expression at the transcriptional or translational level and regulate biological processes, such as epithelial-to-mesenchymal transition, apoptosis, DNA repair and drug efflux, which are tightly associated with drug resistance acquisition in many types of cancer. Interestingly, these ncRNAs are commonly detected in extracellular vesicles (EVs) and are known to be delivered into surrounding cells. This intercellular communication via EVs is currently considered to be important for acquired drug resistance. Here, we review the recent advances in the study of drug resistance in oral cancer by mainly focusing on the function of ncRNAs, since an increasing number of studies have suggested that ncRNAs could be therapeutic targets as well as biomarkers for cancer diagnosis.

## 1. Introduction

Oral cancer is a cancer of the tongue, gums, buccal mucosa, lips, palate and floor of the mouth, and more than 90% of cases are diagnosed as oral squamous cell carcinoma (OSCC) [1]. Among head and neck cancers, oral cancer has a high incidence, and tongue cancer accounts for half of all oral cancers (Figure 1) [2,3,4]. The survival rates and prognoses for advanced and metastatic cancers are poor, and the annual number of deaths related to oral cancer is estimated to be approximately 180,000 [5,6]. In addition, oral cancer has a significant impact on daily life. Surgical resection is commonly used as the first-line treatment for oral cancer, while radiation therapy, chemotherapy, and immunotherapy are used alone, or in combination for cases of advanced or unresectable oral cancer [7,8]. Chemotherapy for oral cancer is commonly based on cisplatin (CDDP), 5-fluorouracil (5-FU), and paclitaxel alone, or in combination. Tumors that are initially sensitive to these drugs often develop resistance to them, which reduces their effectiveness, limits their therapeutic efficacy and is a major cause of recurrence and metastasis. Unfortunately, despite the application of various therapies developed in the past few decades, the five-year overall survival rate of OSCC has remained at approximately 50% [9]. Drug resistance (or chemoresistance) can be divided into two types: intrinsic drug resistance and acquired drug resistance [10,11]. Intrinsic drug resistance refers to the presence of resistance factors in the majority of tumor cells prior to drug administration, while acquired drug resistance refers to resistance that develops due to the expression of resistance factors during drug treatment. Resistance to these anticancer drugs is still a major problem in cancer treatment and will continue to be a major issue in the future.

Previous studies have shown that drug resistance mechanisms that can interfere with therapeutic efficacy are related to altered drug efflux, DNA damage repair, inhibition of apoptosis, mutation of drug targets, epithelial-mesenchymal transition (EMT) and cancer stem cells (CSCs) [12,13,14,15]. A number of these factors are involved in the acquisition of drug resistance, and understanding the mechanisms involved should lead to the discovery of new strategies to improve anticancer efficacy. Recently, evidence has shown that approximately 98% of the human genome is transcribed into RNA that has no protein-coding potential and, hence, is termed non-coding RNA (ncRNA) [16,17]. One kind of ncRNA, microRNA (miRNA), is an endogenous small ncRNA of 18–22 nt in length, which was first discovered in 1993 in the lin-4 locus of *C. elegans* [18]. These small molecules bind to the 3’UTR of the target mRNA in a sequence-specific manner, thereby inhibiting mRNA shedding and translation [19]. In contrast, long non-coding RNA (lncRNA) is a novel class of RNA with a length of greater than 200 nucleotides. The human genome is pervasively transcribed, producing thousands of lncRNAs. Although MALAT1 was well-characterized, most lncRNAs have a low abundance and instability and lack typical signatures of selective constraints [20,21]. In addition, these ncRNAs are encapsulated by the lipid bilayers of extracellular vesicles (EVs) [22] and can exist stably in the bloodstream and transmit biological information to recipient cells. With the discovery and identification of ncRNAs, the functions of dysregulated ncRNAs in OSCC development and drug resistance are gradually being widely recognized [23]. Notably, ncRNAs exhibit both tumor-promoting and tumor-suppressive activity. Here, we summarize the latest evidence associated with drug resistance in OSCC and the underlying mechanisms of cell-to-cell communication via ncRNAs.

## 2. Epithelial-to-Mesenchymal Transition: EMT

Epithelial-to-mesenchymal transition (EMT) is an important biological feature in the developmental process and in wound healing, in which epithelial cells lose their polarity and intercellular tight junctions and acquire mesenchymal-like cell traits accompanied by invasion and migration capabilities [24,25]. In cancer cells, EMT promotes increased tumor-initiating ability and metastatic potential. Furthermore, the phenotypic change of EMT endows increased resistance to several therapeutics [26]. Several EMT signaling pathways have been implicated in drug resistance in cancer cells, and cells that undergo EMT exhibit characteristics similar to those of cancer stem cells (CSCs), such as increased drug efflux pump expression and apoptosis resistance [27]. Therefore, targeting EMT is considered to be a new opportunity to overcome drug resistance in cancer. EMT is a reversible phenomenon and is characterized by downregulation of the epithelial marker E-cadherin and upregulation of the mesenchymal marker vimentin [25]. Various EMT-activating transcription factors (EMT-ATFs) are strongly involved in the regulation of EMT. These factors regulate the entire EMT program in addition to transforming cells into a mesenchymal phenotype, giving cancer cells stem-like properties. These migrating cancer stem cells are a subpopulation of self-renewing tumor cells that can quickly adapt to changes in the surrounding environment. Additionally, EMT is not only important in the development of primary tumors, but also promotes metastasis. Cells that have undergone EMT are more resistant to chemotherapy than other tumor cells, leading to the root cause of recurrence [28,29,30]. These EMT-ATFs, including those in the Snail, TWIST, and ZEB families, bind specifically to the E-cadherin promoter via an E-box and recruit cofactors and histone deacetylases to inhibit E-cadherin transcription [31]. Snail is also well known to be induced via specific pathways, including the TGF-β, Notch, TNF-α, Wnt and hypoxia pathways [32,33,34,35]. Snail also suppresses Raf kinase inhibitory protein (RKIP), an inhibitor of nuclear factor-kappa B (NF-κB) [36]. In addition, Twist plays an important role in EMT by inducing ZEB1 expression in cooperation with Snail. Twist maintains lung cancer cells in a mesenchymal state through the PTEN/PI3K/AKT pathway and inhibits the effects of cisplatin [37].

It has been reported that the expression of let-7d is decreased and that the expression of Twist and Snail is increased in OSCC cell lines. Furthermore, overexpression of let-7 significantly suppresses cisplatin resistance and 5-FU resistance in OSCC cells, suggesting that let-7 negatively regulates EMT and is involved in the regulation of chemotherapeutic resistance [38]. Knockdown of ZEB1 and Snail expression via small interfering RNAs (siRNAs) can improve the sensitivity of cancer cells to chemotherapy. Therefore, understanding the regulatory mechanisms of the ZEB1 machinery is important in overcoming drug resistance in oral cancer [39]. ZEB1 is regulated by several ncRNAs, such as those in the miR-200 family [40,41,42,43,44,45,46]. miR-200 family members directly bind to ZEB1 and ZEB2 mRNAs in many types of cancers, including oral cancer, and induce EMT by decreasing ZEB1 and ZEB2 expression. Additionally, the sequences of miR-200 family members, such as miR-200a, miR-200b and the related miR-429, are located within a 7.5-kb polycistronic primary miRNA transcript [44]. In epithelial cells, these miR-200 family members can be expressed and downregulate the ZEB1/SIP1 complex. Conversely, in mesenchymal cells, ZEB1 and SIP1 suppress the expression of miR-200 family members through a conserved pair of ZEB-type E-box elements located proximal to the transcription start site. These double-negative feedback loops play a pivotal role in controlling ZEB1/SIP1 and miR-200 family expression, which regulates the cellular phenotype and has direct relevance to the roles of these factors in tumor progression. Furthermore, hypermethylation of the ZEB1 and ZEB2 promoters plays an important role in the regulation of EMT during carcinogenesis. The same ncRNA-mediated EMT-inducing mechanism has been observed in oral cancer [42,46,47]. Miyazaki et al. reported that CD44s induced EMT by repressing miR-200c and upregulating ZEB1 [47].

In addition to regulation by miRNAs, regulation of EMT via lncRNAs has been extensively reported. Lu et al. showed that the lncRNA HOTAIR promoted the cancer stemness and metastasis of oral carcinoma stem cells through modulation of EMT. Silence of HOTAIR in oral carcinoma stem cells significantly inhibited their cancer stemness, invasiveness and tumorigenicity in xenograft mouse models. By contrast, overexpression of HOTAIR in OSCC enhanced their metastatic potential and EMT characteristics [48]. As another example of the ncRNA-mediated regulation of ZEB1 in mesenchymal breast tumor cells, lncRNA-PNUTS serves as a competitive sponge for miR-205 during EMT [49]. The expression of lncRNA-PNUTS is elevated and correlates with the mRNA level of ZEB1. A number of lncRNAs have been identified as EMT modulators in oral cancer [48,49,50,51,52,53,54,55,56,57,58,59,60,61,62,63,64,65,66,67,68,69,70,71,72,73,74,75,76,77,78,79,80,81,82,83,84,85,86,87,88,89,90,91,92,93,94,95,96,97,98,99,100,101,102,103,104,105,106,107,108,109]. The details of these lncRNAs are summarized in Table 1. Thus, the expression levels of these EMT-ATFs are regulated by many ncRNAs, including miRNAs and lncRNAs, resulting in EMT, and these ncRNAs are subsequently involved in the development of drug resistance.

## 3. Apoptosis

Apoptosis is a programmed form of cell death that is inherent in cells and has been reported to be important in tumorigenesis and the chemotherapeutic response. Any change in the molecules or pathways associated with apoptosis results in inhibition of apoptosis and resistance to chemotherapeutic agents [110]. Apoptosis is highly regulated by Bcl-2 family members, which consist of antiapoptotic and proapoptotic proteins. Previous studies have reported that Bcl-2, Bcl-XL, and Mcl-1, which are antiapoptotic proteins, are overexpressed in cisplatin-resistant OSCC cell lines [111]. Interestingly, it is possible to promote apoptosis by downregulating the expression of these proteins; Bcl-XL can promote apoptosis by regulating upstream regulators [112,113]. Mcl-1 is regulated by STAT3 and AKT-mediated GSK3b signaling, and chemical inhibitors have been shown to be able to restore the apoptotic process [114]. Similar to these proteins, proapoptotic proteins are also known to be involved in chemotherapeutic resistance, and Bax has been shown to be downregulated in cancer tissues from cisplatin-resistant OSCC patients and in cisplatin-resistant OSCC cell lines. Bax is a downstream gene in the p53/AKT pathway, and inhibition of Akt promotes apoptosis [115,116]. Inhibitors of apoptosis (IAPs) are known to have a baculoviral IAP repeat (BIR) domain, comprised of 70–80 amino acids, that binds to and inhibits caspases, thereby suppressing cell death. X-linked inhibitor of apoptosis (XIAP) is an IAP that is a negative regulator of apoptosis and has been shown to antagonize the apoptotic cascade by directly inhibiting caspases [117]. Survivin is also a member of the IAP family and has been implicated in chemoresistance by its inhibitory effect on apoptosis mediated through binding to caspases. In previous studies, it has been reported that survivin expression is upregulated in HNSCC and is positively correlated with malignant characteristics, suggesting that it could be a biomarker for cancer [118]. Notably, p53 is a tumor suppressor, and mutated p53 is unable to induce apoptosis and is involved in chemotherapeutic resistance. Clinical evidence suggests that the prevalence of p53-mutated tumors is higher in patients who fail to respond to neoadjuvant therapy than in those who do respond [119]. As described above, changes in the expression of apoptotic proteins and proapoptotic proteins result in drug resistance in cancer cells.

A number of ncRNAs are involved in the regulation of these proteins to permit escape from the effects of anticancer drugs. The expression of miR-155 is elevated in OSCC patient-derived tissues [120]. This upregulated miR-155 directly binds to FOXO3a, which is a well-defined tumor suppressor gene. The expression of miR-155 was significantly higher but FOXO3a mRNA and protein expression was obviously lower in cisplatin-resistant oral cancer cell lines than in the parental cells. Inhibition of miR-155 expression upregulated FOXO3a expression, suppressed proliferation, promoted apoptosis and enhanced cisplatin sensitivity in oral cancer cells. FOXO3a is also regulated by miR-223 [121].

The miR-371/372/373 miRNA cluster is located on human chromosome 19q13.4, which is frequently acquired in head and neck cancers [122]. Additionally, expression of miR-372 and miR-373 is correlated with poor prognosis in OSCC patients. Simultaneous silencing of miR-371/372/373 in SAS cells was found to decrease their oncogenic potential, increase their cisplatin sensitivity, activate p53 and upregulate the expression of Bad and DKK1. In contrast, upregulation of the miR-371/372/373 cluster slightly enhanced the expression of AKT, β-catenin and Src, leading to promotion of their oncogenic effects and drug resistance. Additionally, miR-372 regulates zinc finger and BTB domain containing 7A (ZBTB7A), which is a transcriptional regulator and is involved in regulating a great diversity of physiological and oncogenic processes [123]. ZBTB7A modulates the expression of the death receptors TRAIL-R1, TRAIL-R2, Fas and p53. The ZBTB7A-TRAIL-R2 cascade is involved in both the extrinsic and intrinsic cisplatin-induced apoptosis-related pathways. MMP2 expression was found to be higher in OSCC tissues than in adjacent tissues, but miR-29a expression was lower in OSCC tissues than in adjacent tissues. Exogenous overexpression of miR-29a significantly inhibited OSCC cell invasion and apoptosis resistance by inhibiting MMP2 directly [124]. The long non-coding RNA urothelial cancer associated 1 (UCA1) is generally regarded as an oncogene in some cancers [125]. UCA1 expression was found to be upregulated in OSCC cells, which facilitated proliferation, enhanced cisplatin chemoresistance, and suppressed apoptosis. UCA1 could interact with miR-184 to repress its expression. Downregulation of miR-184 partially reversed the tumor-suppressive and cisplatin chemosensitizing effects of UCA1 knockdown in cisplatin-resistant OSCC cells. UCA1 acts as a miR-184 sponge, and downregulation of miR-184 then modulates the expression of SF1. Modulation of SF1 accelerates cancer cell proliferation, increases cisplatin chemoresistance and partially suppresses apoptosis. The expression of the lncRNA CEBPA-DT/CEBPA/BCL2 is upregulated in cisplatin-resistant OSCC cells compared with the parental cells [126]. Downregulation of CEBPA-DT enhances cisplatin sensitivity and facilitates apoptosis in cisplatin-resistant OSCC cells. In addition, CEBPA-DT regulates cisplatin chemosensitivity through CEBPA/BCL2-mediated apoptosis.

In tongue squamous cell carcinoma (TSCC), miRNA processing-related lncRNA (MPRL) is significantly upregulated in TSCC cell lines treated with cisplatin and transactivated by E2F1 [127]. MPRL controls mitochondrial fission and cisplatin sensitivity through miR-483-5p. Cytoplasmic MPRL directly binds to pre-miR-483 within the loop region and inhibits pre-miR-483 recognition and cleavage by the TRBP-DICER complex, leading to downregulated generation of miR-483-5p and upregulated expression of the miR-483-5p downstream target FIS1. High expression of MPRL and pre-miR-483 and low expression of miR-483-5p were found to be significantly associated with chemosensitivity and better prognosis in TSCC patients. In addition to these miR-483-5p and MPRL, Rin Z et al. showed that upregulation of CILA1 promotes EMT phenotype, invasiveness and chemo-resistance in TSCC cells, whereas the inhibition of CILA1 expression induces mesenchymal-epithelial transition (MET) and chemo-sensitivity [93]. The expression level of CILA1 is correlated with poor prognosis in TSCC patients. It indicates that ncRNAs, which up-regulated in cancer patients, are also useful as biomarkers.

Moreover, not only cancer cells, but also cells within the cancer microenvironment are regulated by lncRNAs and are involved in drug resistance [128]. Midkine (MK) is a heparin-binding growth factor that promotes carcinogenesis and chemoresistance. MK induces cisplatin resistance in OSCC cells. Cancer-associated fibroblasts (CAFs) secrete increased levels of MK, which abrogates cisplatin-induced cell death. Moreover, MK increases the expression of the lncRNA ANRIL in tumor cells. lncRNA ANRIL knockdown in tumor cells inhibits proliferation, induces apoptosis and increases cisplatin cytotoxicity in tumor cells via impairment of the drug transporters MRP1 and ABCC2, which can be restored by treatment with human MK in a caspase-3/BCL-2-dependent manner. Other apoptosis-related lncRNAs are summarized in Table 2: XIST [129], HOXA11-AS [130], KCNQ1OT1 [131,132], CASC2 [133,134], HOTAIR [135]. Thus, targeting lncRNAs, not only in cancer cells, but also in cells in the surrounding cancer microenvironment, may be a new therapeutic approach for OSCC.

## 4. DNA Damage and Repair

When DNA damage occurs, an intrinsic DNA damage response mechanism normally suppresses or completely disrupts DNA synthesis, inhibits cell cycle progression and activates the DNA repair pathway [136]. Abnormalities in the DNA repair pathway have been linked to cancer development and progression, and chemotherapeutic agents induce cell death by directly or indirectly causing DNA damage. However, cancer cells are known to reduce their susceptibility to chemotherapeutic agents by enhancing their DNA repair ability or preventing the induction of cell death [137]. DNA damage repair is mainly carried out via mismatch repair (MMR), nucleotide excision repair (NER), base excision repair (BER), interstrand crosslink repair (ICR), homologous recombination (HR) repair, translesion DNA synthesis (TLS) and nonhomologous end joining (NHEJ) [138,139]. Previous studies have reported that alterations in genes and pathways associated with NER are associated with cisplatin-resistant phenotypes in many cancers, including gastric and ovarian cancers [140]. Cisplatin-resistant TSCC is known to exhibit increased expression of excision repair cross complementation group 1 (ERCC1), a key component of the NER pathway, and previous studies have shown that silencing ERCC1 reverses resistance to cisplatin in gastric and ovarian cancers. Studies have shown that silencing ERCC1 reverses resistance to cisplatin in gastric and ovarian cancers [141]. Furthermore, it was reported that patients with low levels of ERCC1 who received cisplatin chemotherapy had a statistically significant improvement in survival compared to patients with high levels of ERCC1 who received cisplatin chemotherapy [142]. HR is also known to be a pathway that can preserve the entire DNA sequence. DNA damage caused by chemotherapeutic agents has been shown to lead to the development of drug resistance through DNA repair by RAD51, an HR repair protein that is important for identifying homologous sequences on sister chromatids [143]. It has been reported that BRCA1, or breast cancer type 1 protein, which is involved in cell cycle checkpoint activation, is also involved in DNA double-strand break (DSB) repair. It has been reported that BRCA1 is an important regulator of RAD51 function. Cancers lacking BRCA1 are susceptible to cisplatin because of the accumulation of DSBs due to the inability of DNA repair by HR. However, it has been reported that cancers with BRCA1 mutation acquire resistance to cisplatin due to secondary mutations that restore BRCA1 function [144]. Similar to HR repair, DSBs can also be repaired by NHEJ, which plays an important role in functional genomic reconstruction after DNA damage. TRIP13 is a member of the AAA + ATPase family of proteins and is known to bind to repair proteins such as KU70, KU80, and DNA-PKcs [145]. During NHEJ, KU70 and KU80 are activated by recruiting DNA-PKcs, and XRCC4 and artemins are then incorporated to form the DNA-PK complex. These results suggest that TRIP13 may provide the energy necessary for the formation of the DNA-PK complex and that TRIP13 may be involved in cell survival by promoting NHEJ. It has also been reported that head and neck squamous cell carcinoma (HNSCC) cells overexpressing TRIP13 are resistant to chemotherapy and radiation.

Thus, a variety of genes are associated with DNA repair, resulting in drug resistance. These genes are also regulated by some ncRNAs in oral cancer. Moreover, BRD4, a member of the bromodomain and extra-terminal domain (BET) protein family, plays a pivotal role in promoting DNA repair as well as regulating super-enhancers and transcriptional activation of oncogenes in cancer. The BRD4 gene is recognized as a promising target for cancer therapy. Notably, miRNA mimics can simultaneously target several tumor-promoting genes, and BRD4 may be useful as a therapeutic target for tumor-suppressive miRNAs [146]. By screening more than 2000 miRNA mimics, miR-1293, miR-876-3p, and miR-6571-5p were identified as tumor-suppressive miRNAs targeting BRD4. Notably, miR-1293 also suppressed DNA repair pathways by directly suppressing the DNA repair genes APEX1 (apurinic-apyrimidinic endonuclease 1), RPA1 (replication protein A1) and POLD4 (DNA polymerase delta 4, accessory subunit). Simultaneous suppression of BRD4 and these DNA repair genes synergistically inhibited tumor cell growth. Vomeronasal type-1 receptor 5 (VN1R5), which is associated with cisplatin resistance, is highly expressed in cisplatin-resistant HNSCC cells and tissues [147]. Expression of the lncRNA lnc-POP1-1 is transcriptionally regulated by VN1R5, which activates the transcription factor specificity protein 1 (Sp1) via the cyclic AMP (cAMP)/protein kinase A (PKA) pathway. lnc-POP1-1 was found to bind directly to the minichromosome maintenance deficient 5 (MCM5) protein and inhibit the ubiquitination of the MCM5 protein, thereby slowing its degradation, then facilitating the repair of DNA damage caused by cisplatin. Although it is still unclear which lncRNAs are involved in DNA repair in oral cancer, the involvement of lncRNAs in EMT and other drug resistance-related phenotypes suggests that many lncRNAs may also be involved in DNA repair.

## 5. Stem Cells and Changes in Drug Efflux (Drug Transporters)

Cancer stem cells (CSCs), also known as tumor-initiating cells (TICs), have been reported to exist in a variety of cancers, including head and neck cancers [148]. CSCs are a subpopulation of tumor cells with self-renewal and tumorigenic properties that divide unequally to form heterogeneous cell populations, which often transform into invasive and metastatic types of tumors and are thought to be the cause of recurrence and metastasis [149,150]. They are also capable of inducing cell cycle arrest and are considered to possess the ability to confer resistance to chemotherapeutic agents and radiotherapy [151]. Although the developmental origin of CSCs is still elusive, it has been suggested that the accumulation of genetic mutations in normal stem cells may result in enhanced self-renewal and transformation, or that mutations in progenitor cells or mature cells may result in dedifferentiation, leading to the acquisition of self-renewal properties and tumorigenicity [152,153]. CSCs may acquire resistance to chemotherapy via several mechanisms: overexpression of ATP-binding cassette (ABC) transporters and antiapoptotic molecules, abnormal activation of signaling pathways, acquisition of multidrug resistance due to enhanced DNA repair activity, and the presence of a tumor microenvironment (TME) that protects tumor cells from drugs. There is a wide range of possibilities, such as the formation of a niche to protect tumor cells from drugs via the tumor microenvironment (TME) or dormancy (inactivity) in the body, followed by activation (after several years or decades) [154,155]. In the tumor microenvironment surrounding cells, mesenchymal stem cells (MSCs) could affect cancer cells to enhance their stemness [156,157,158]. Cancer cells co-cultured with MSCs exhibited upregulated expression of the lncRNA AGAP2-AS1, which enhanced cancer stemness [158]. AGAP2-AS1 forms a complex with HuR, and this complex binds to CPT1, leading to RNA stability. In addition, AGAP2-AS1 acts as miR-15a-5p sponge.

One of the main causes of chemotherapeutic resistance in CSCs is an increase in the expression of genes encoding drug efflux pumps, which promotes the efflux of anticancer drugs from tumor cells and decreases the intracellular drug concentration, leading to multidrug resistance. Among the transporters involved in these changes, the adenosine phosphate (ATP)-binding cassette (ABC) transporter family is well known. All ABC transporters are composed of two nucleotide-binding domains (NBDs) and two transmembrane domains (TMDs) and are responsible for transporting a wide range of substrates, such as amino acids, ions, sugars, lipids, and drugs [159]. TMDs play a role in recognizing substrates and mediating their passage through the cell membrane, and the diversity of TMDs indicates differences in substrate specificity. The ABC transporter genes are classified into seven subfamilies (A to G). Among these transporters, ABCB1 (P-glycoprotein/P-gp/MDR1), ABCC1 (multidrug resistance protein 1/MDR1), and ABCG2 (breast cancer resistance protein/BCRP) have been extensively reported in many studies [160]. The most common indicator of multidrug resistance is P-glycoprotein (P-gp), which is a product of the multidrug resistance gene (MDR1) and a 170 kDa transmembrane phosphoglycoprotein that functions as an ATP-dependent efflux pump [161]. Notably, P-gp may eliminate paclitaxel and vincristine from P-gp-positive OSCC cells. Additionally, taxanes, alkaloids, podophyllotoxins, and other molecules of plant origin are the main substrates of P-gp in cancer cells [162]. In one study, P-gp expression in OSCC was predominantly increased in recurrent tumors compared to primary tumors, and P-gp expression was significantly correlated with the severity of dysplasia. These results suggest that the expression level of P-gp may be a prognostic indicator in patients with OSCC [163]. Furthermore, ABCC1 is also a member of the ABC subfamily and acts on organic anions such as glutathione, glucuronic acid, and drugs bound to sulfate as substrates [164]. It is known to be upregulated in OSCC cell lines treated with vincristine or CDDP and is correlated with chemoresistance and poor prognosis. The third drug transporter is BCRP, which is encoded by ABCG2. It has been shown that the expression of ABCG2 is increased in 5-FU- and CDDP-resistant OSCC cells compared to OSCC cells that are sensitive to these drugs [165]. Notably, BCRP may also be a marker for stem cell-rich side population (SP) cells, which have been reported to be tumorigenic, express stem-like genes, and be resistant to chemotherapy. In OSCC, SP cells were found to express higher levels of ABCG2 than non-SP cells, suggesting that high expression of BCRP in SP cells promotes the progression of drug resistance, proliferation and tumor invasion [166]. The expression of the lncRNA LINC00963 was found to be upregulated in oral cancer tissues, and knockdown of this lncRNA led to a decrease in stemness via suppression of ABCG2 [167].

Therefore, although few studies have shown a relationship between ncRNAs and CSCs in oral cancer, the development of therapies that target CSCs could be considered to mitigate chemoresistance.

## 6. Other ncRNAs: circRNAs and piRNAs in Drug Resistance

In addition to miRNAs and lncRNAs, a number of ncRNAs such as circular RNAs (circRNAs) and PIWI-interacting RNAs (piRNAs) have been reported. Of note, circRNAs are characterized by closed loop structures without a polyadenylated tail, leading to the under-estimation of the existence of circular RNAs in previous polyadenylated transcriptome analyses [168]. One of the most important mechanisms of circRNAs is their function as a sponge of miRNAs [168,169,170]. Furthermore, circRNAs strongly inhibit their target miRNAs, which leads an increase in the expression level of the mRNA that miRNAs target. Accumulating evidence suggests that circRNAs are also involved in drug resistance in cancer [171,172,173,174,175]. The expression level of circRNA_0025202 is downregulated in tamoxifen-resistant MCF-7 cells [171]. A large cohort of breast cancer showed that low expression of circRNA_0025202 is correlated with lymphatic metastasis and histological grade. In normal tissue, circRNA_0025202 acts as a sponge for miR-182-5p. Downregulation of miR-182-5p induces the expression and activity of FOXO3a, which enhances tumor inhibition and tamoxifen sensitization effects. In gastric cancer patients, the expression of circCUL2 was negatively regulated [172]. One of the targets of circCUL2 is miR-142-3p; this miRNA induces autophagy via inhibition of ROCK2. As another function of circRNAs, circRNA-SORE is upregulated in sorafenib-resistant hepatocellular carcinoma cells and its depletion substantially increases sorafenib sensitivity [173]. Additionally, circRNA-SORE binds the master oncogenic protein YBX1 in the cytoplasm, which prevents YBX1 degradation. Thus, circRNA-SORE contributed to stabilization of YBX1 protein. Notably, circRNAs also play an important role in the OSCC [176,177,178,179,180,181] and circ_0000140 was downregulated in OSCC [176]. Furthermore, circ_0000140 acts as a sponge for miR-31, then, upregulates its target gene LATS2, which is associated with EMT. Significantly, circIGHG/miR-142-5p/IGF2BP3 axis, and circEPSTI1/mir-942-5p/LTBP2 axis, circUHRF1/miR-526b-5p/c-Myc/TGF-β1/ESRP1 axis were also shown to promote EMT [177,178,179]. Furthermore, circ_0001461/miR-145/TLR4/NF-kB axis promotes the resistance of TNF-a-induced apoptosis [181].

Moreover, piRNAs constitute small ncRNA molecules of approximately 24–31 nt in length and bind to the PIWI protein family to regulate many pathways at the transcriptional or post-transcriptional level [182,183]. Notably, piRNAs are mostly expressed in the germline and their main function is transposon silencing, although their function in mammalian somatic tissues is not so clear. Aberrant expression of piRNAs in cancer cells has been detected and is expected to be used as a biomarker; however, little is known about the function of piRNAs in cancer cells [184,185,186,187]. For instance, piRNA-30473 is involved in the upregulation of WTAP, an m6A mRNA methylase. An increased m6A level contributes to the tumorigenesis in diffuse large B-cell lymphoma [184]. The involvement of piRNAs in drug resistance has also been reported [185,186]. Doxorubicin (DOX) resistant fibrosarcoma cells express piR-39980. In DOX sensitive cells, inhibition of piR-39980 induces DOX resistance by suppressing RRM2 and CYP1A2 [185]. RRM2 enhances DNA repair in DOX resistant cells, while CYP1A2 decreases intracellular DOX accumulation. Another example of drug resistance associated piRNA is that the expression level of piRNA-36712 is low in breast cancer [186]. Furthermore, piRNA-36712 is associated with the downregulation of SEPW1 in normal breast tissue. In cancer cells, downregulation of piRNA-36712 mediates the upregulation of SEPW1, and then suppresses p53 and promotes EMT. In addition, synergistic anticancer effects with the paclitaxel and DOX was observed when piRNA-36712 was used in combination. In this way, understanding the function of piRNAs is also important to overcome the drug resistance. Several piRNAs have been reported to be involved in OSCC [187], but no reports existed on drug resistance. Thus, future analysis of piRNAs in OSCC will be of great value.

## 7. Extracellular Vesicles: EVs

Given that the TME is a critical factor in increased drug resistance, understanding cell-to-cell communication through ncRNAs is important for understanding drug resistance. Exosomes are classified as small extracellular vesicles (sEVs) with a diameter ranging from 30 to 150 nm (average, 100 nm) and are of endosomal origin. EVs, including exosomes, are secreted by most types of cells. EVs are enclosed by a lipid bilayer and carry various biomolecules, such as proteins, glycans, lipids, metabolites, RNA and DNA [188]. When EVs are taken up by other cells, these cargoes are transferred and influence the phenotype of the recipient cells. Thus, EVs are appreciated as essential mediators of cell-to-cell communication. In the field of cancer biology, EVs have been found to play crucial roles in cancer metastasis [189] and drug resistance [190,191,192,193,194,195]. In oral cancer, EVs also mediate drug resistance by transferring their cargoes to recipient cells [196,197,198,199,200,201,202].

As described above, miRNAs are important for drug resistance in oral cancer. Considering that many miRNAs are contained in EVs, it is very important to understand cell-to-cell communication via EVs. Kirave et al. showed that the expression of miR-155 was upregulated in OSCC cells, and miR-155 was also highly detected in EVs [203]. This exosomal miR-155, derived from cisplatin-resistant cells, transformed cisplatin-sensitive cells to cisplatin-resistant cells. In contrast, downregulation of exosomal miR-30a was found to be involved in OSCC recurrence [204]. In this study, miR-30a was found to directly bind to Beclin1, which is associated with autophagy. EVs derived from cisplatin-resistant cells with high miR-30a expression decreased the Beclin1 level. More importantly, these EVs resensitized the resistant cells to cisplatin. These results indicate that exosomal miR-30a suppressed OSCC, although cancer cells could sometimes escape the effects of tumor-suppressive miRNAs. Additionally, miRNAs encapsulated in EVs can also act as sponges for miRNAs [205]. Circ-SCMH1 was found to be upregulated in cisplatin-resistant OSCC tissues and cells; circ-SCMH1 was an EV cargo, and cisplatin-resistant OSCC cell-derived EVs facilitated circ-SCMH1 upregulation in the parental cells. Circ-SCMH1 directly sponged miR-338-3p and indirectly regulated LIN28B, a target gene of miR-338-3p. Exosomal lncRNA APCDD1L-AS1 conferred resistance to 5-FU via the miR-1224-5p/NSD2 axis in OSCC [206]. The expression of the lncRNA ZFAS1 attenuated miR-421 expression in OSCC, leading to the acquisition of cisplatin resistance [207].

As EVs play a pivotal role in cancer drug resistance, EV-targeted therapies are currently being developed. Khoo et al. showed that inhibition of EV secretion from cisplatin-resistant cells restored cisplatin sensitivity [208]. Epidermal growth factor receptor (EGFR) signaling is frequently increased in OSCC; thus EGFR is molecularly targeted by the therapeutic antibody cetuximab [209]. Cetuximab has only a partial effect; thus, some cancer cells survive. The surviving cells then secrete more EGFR-expressing EVs than normal cells. These EGFR-expressing EVs act as decoys for cetuximab, which causes cetuximab resistance. Thus, it is important to consider not only cancer cells but also EVs secreted by cancer cells when considering therapeutics. Furthermore, EVs from tumor microenvironment cells, including CAFs, also affect cancer drug resistance [196,210]. In head and neck cancer, exosomal miR-196a derived from CAFs was found to regulate CDKN1B and ING5 [211]. Depletion of exosomal miR-196a restored cisplatin sensitivity in head and neck cancer cells. In contrast, normal cell-derived EVs have the potential to suppress tumor progression [212,213,214,215]. Notably, miR-200c was packaged into EVs derived from normal tongue epithelial cells (NTECs), which were transferred into docetaxel-resistant cells, and the transferred miR-200c influenced docetaxel resistance by inhibiting TUBB3 and PPP2R1B [212]. Therefore, EV-mediated cell-to-cell communication could be a potential therapeutic option (Figure 2).

## 8. Conclusions

Understanding the molecular mechanisms of drug resistance is critical for improving patient prognosis in oral cancer, consequently, a large number of studies have been conducted worldwide. Since miRNAs are well known to be associated with cancer biology and are easy to analyze, researchers have intensively and extensively focused on investigating miRNAs for the past two decades. Conversely, due to their low abundance and difficulty of analysis, examination of lncRNAs has been delayed; however, recent technological advances in next-generation sequencing have paved the way for investigating lncRNAs in cancer. As described above, ncRNAs are involved in processes related to drug resistance acquisition, such EMT, apoptosis, DNA repair, and CSC development (Figure 3). Therefore, targeting both ncRNAs themselves and their signaling targets could be a better therapeutic strategy for preventing or overcoming drug resistance.

Moreover, ncRNAs, particularly miRNAs, have been identified as being packaged into EVs. Notably, ncRNAs packaged as EVs function as natural intercellular communication tools. As they can be readily detected in body fluids, such as serum, plasma, saliva and urine, EVs have attracted considerable attention as liquid biopsy biomarkers. This kind of biomarker could be used in a precision medicine approach, not only to detect cancer at an early stage, but also to predict the responsiveness to anticancer drugs, the acquisition of drug resistance, and the possibility of recurrence.

## Figures and Tables

**Figure 1 biomolecules-12-00284-f001:**
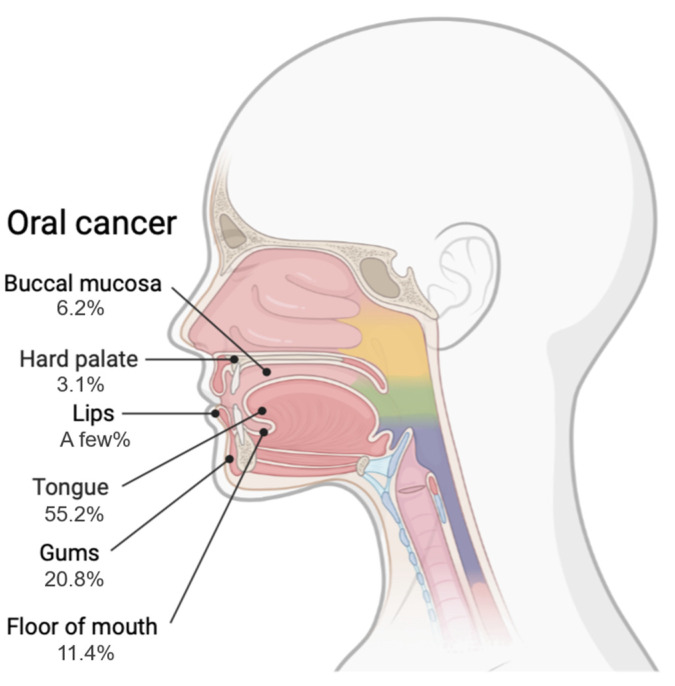
Sites of Oral Cancer and Frequency of Occurrence. Tongue cancer: Cancer that occurs in the front two-thirds of the tongue (the movable part of the tongue). Gingival cancer: Cancer that forms in the mucous membrane of the dead flesh. Cancer of the lip and buccal mucosa: Cancer that forms on the inside of the lips and cheeks. Cancer of the hard palate: Cancer that forms in the mucous membrane of the roof of the mouth.

**Figure 2 biomolecules-12-00284-f002:**
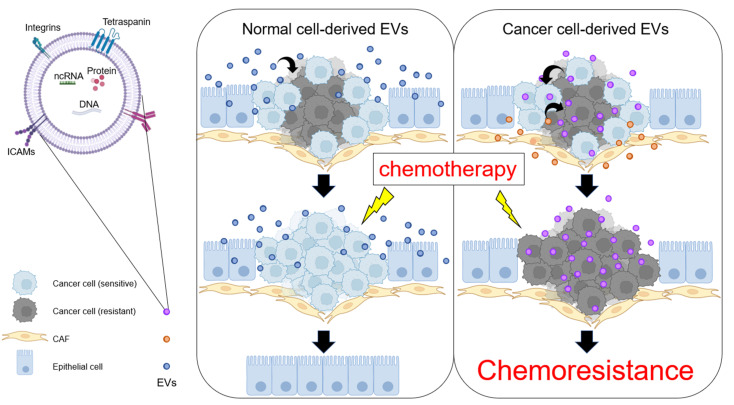
An overview of EV-mediated chemotherapy resistance. EVs derived from normal cells have the potential to inhibit tumor progression. In contrast, EVs derived from chemoresistant cells promote the conversion of chemosensitive cells into chemoresistant cells.

**Figure 3 biomolecules-12-00284-f003:**
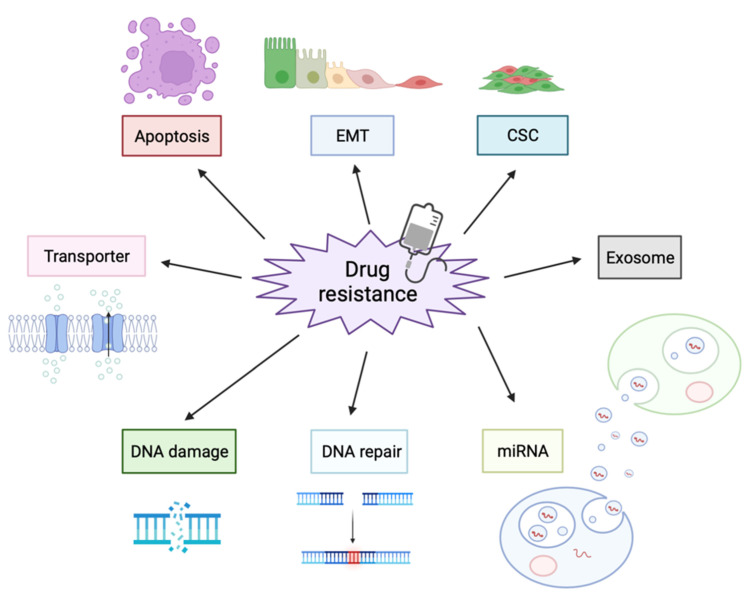
Resistance Mechanisms to Anticancer Drugs in Oral Cancer. Cancer cells use a variety of mechanisms to reduce the efficacy of anticancer drugs, such as increasing transporter, inducing EMT, emitting anticancer drugs in EVs, and DNA repairing.

**Table 1 biomolecules-12-00284-t001:** EMT inducing lncRNAs in OSCC.

lncRNAs	Expression	Target and (Function)	Cancer Type	Reference
HOTAIR	up	(upregulation of the cancer stemness, invasiveness and tumorigenicity)	OSCC	[48]
PNUTS	up	miR-205	OSCC	[49]
LINC01303	up	miR-429	OSCC	[50]
LINC00460	up	(PRDX1 promoted the transcription of LINC00460)	HNSCC	[51]
		miR-4443	HNSCC	[52]
		miR-320b/IGF2BP	TSCC	[53]
MALAT1	up	(upregulation of P-glycoprotein)	OSCC	[54]
		(downregulation of MMP-9)	TSCC	[55]
		miR-101/EZH2	OSCC	[56]
		miR-125b/STAT3	OSCC	[57]
PVT1	up	miR-194-5p/HIF1a	OSCC	[58]
		miR-150-5p/GLUT-1	OSCC	[59]
CYTOR	up	miR-1252-5p and miR-3148	OSCC	[60]
NEAT1	up	miR-365/RGS20	OSCC	[61]
HOXA11-AS	up	miR-98-5p/YBX2	OSCC	[62]
		miR-518a-3p/PDK1	OSCC	[63]
LOLA1	up	(activation of the AKT/GSK-3β pathway.)	OSCC	[64]
AC007271.3		miR-125b-2-3p/Slug/E-cadherin	OSCC	[65]
LINC01296	up	SRSF1	OSCC	[66]
ADAMTS9-AS2	up	miR-600/EZH2	OSCC	[67]
SNHG16	up	miR-17-5p/CCND1	OSCC	[68]
		miR-302b-3p/SLC2A4	OSCC	[69]
AATBC	up	miR-1237-3p/PNN/ZEB1	nasopharyngeal carcinoma	[70]
MIR4435-2HG	up	miR-296-5p/Akt2/SNAI1	OSCC	[71]
UCA1	up	miR-124/JAG1	TSCC	[72]
PART1	down	miR-503-5p	TSCC	[73]
LINC00319	up	miR-199a-5p/FZD4	OSCC	[74]
TIRY	up	miR-14	OSCC	[75]
ZNF667-AS1	up	(inhibition of TGF-β1)	OSCC	[76]
SNHG12	up	SNHG12/miR-326/E2F1	OSCC	[77]
HOXC13-AS	up	miR-378g	OSCC	[78]
PRKG1-AS1	up	(acceleration of the cell growth, invasion, and migration)	OSCC	[79]
AFAP1-AS1	up	miR-145/HOXA1	OSCC	[80]
HAS2-AS1	up	HIF-1α	hypoxic OSCC	[81]
CRNDE	up	(activation of the Wnt/β-catenin signaling pathway)	OSCC	[82]
MYOSLID	up	(suppression of the Slug, PDPN and LAMB3 expression)	OSCC	[83]
LINC02487	down	USP17/SNAI1	OSCC	[84]
LINC01116	up	miR-136/FN1	OSCC	[85]
FOXD1-AS1	up	miR-369-3p/FOXD1	HNSCC	[86]
KRT16P3	up	(modulation of JAK2/STAT3 signaling pathway)	TSCC	[87]
MEG3	down	miR-421/E-cadherin	HNSCC	[88]
		miR-21	OSCC	[89]
		miR-548d-3p/SOCS5/SOCS6	OSCC	[90]
HCP5	up	miR-140-5p/SOX4	OSCC	[91]
ZEB1-AS1	up	miR-23a-3p	OSCC	[92]
CILA1	up	(activation of the Wnt/β-catenin signaling pathway)	TSCC	[93]
LINC00958	up	miR-627-5p/YBX2	OSCC	[94]
		miR-185-5p/YWHAZ	OSCC	[95]
		miR-211-5p/CENPK	TSCC	[96]
IGFL2-AS1	up	miR-1224-5p/SATB1	TSCC	[97]
H19	up	miR-675-5p/PFKFB3	OSCC	[98]
HNF1A-AS1	up	(activation of the Notch signaling pathway)	OSCC	[99]
LINC00664	up	miR-411-5p/KLF9	OSCC	[100]
HOTTIP	up	(downregulation of the Bcl-2, upregulation of the Bax)	TSCC	[101]
		miR-124-3p/HMGA2	TSCC	[102]
AC132217.4	up	KLF8/AC132217.4/IGF2	OSCC	[103]
ZEB2-AS1	up	(activation of TGF-β1-induced EMT)	HNSCC	[104]
MIAT	up	(activation of Wnt/β-catenin signaling pathway)	TSCC	[105]
NKILA	down	(inhibition of the NF-κB signaling pathway)	OSCC	[106]
SLC16A1-AS1	up	(inhibition of G0/G1 cell cycle arrest)	OSCC	[107]
PTTG3P	up	miR-142-5p/JAG1	TSCC	[108]
FOXC2-AS1	up	miR-6868-5p/E2F3	TSCC	[109]

**Table 2 biomolecules-12-00284-t002:** lncRNAs associated with anti-apoptosis in OSCC.

lncRNAs	Expression	Target and (Function)	Cancer Type	Reference
UCA1	up	miR-184/SF1	OSCC	[125]
CEBPA-DT	up	CEBPA/BCL2	OSCC	[126]
MPRL	up	miR-483-5p/FIS1	TSCC	[127]
ANRIL	up	(Midkine increased the expression of lncRNA ANRIL in the tumor cells)	CAFs with OSCC	[128]
XIST	up	miR-27b-3p	OSCC	[129]
HOXA11-AS	up	miR-214-3p/PIM1	OSCC	[130]
KCNQ1OT1	up	miR-211-5p/Ezrin	TSCC	[131]
		miR-185-5p/Rab14	OSCC	[132]
CASC2	down	miR-21/PDCD4	OSCC	[133]
		miR-31-5p/KANK1	OSCC	[134]
HOTAIR	up	(the downregulation of the expression of MAP1LC3B, beclin1, ATG3 and ATG7)	OSCC	[135]

## Data Availability

Not applicable.

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
