# Peer review of "Impact of Non-Coding RNAs on Chemotherapeutic Resistance in Oral Cancer"

_biomolecules, 2022, doi:10.3390/biom12020284_

Round 1

Reviewer 1 Report

Yamaguchi et.al. reviewed the function of ncRNA on drug resistance in oral cancer. They also described the ncRNAs packaged in extracellular vesicles and their role in cancer drug resistance. The work is clearly presented showing lncRNAs and miRNAs regulating EMT, apoptosis, cancer stem cells, DNA damage and repair. The reviewer has two minor comments:

  • Please check the spelling throughout the manuscript. For example, Figure 3, Drug resistance
  • Lines 507-512: this paragraph does not make sense at the end of the manuscript. A paragraph to conclude the manuscript should be added.

Author Response

Point-by-point responses to each of the reviewers’ comments

We are grateful to all the reviewers for their critical comments and insightful suggestions that have helped us considerably improve our paper. As indicated in the responses that follow, we have taken all of these comments and suggestions into account in the revised version of our paper. Changes in the revised manuscript are highlighted in red color.

Reviewer1

Yamaguchi et.al. reviewed the function of ncRNA on drug resistance in oral cancer. They also described the ncRNAs packaged in extracellular vesicles and their role in cancer drug resistance. The work is clearly presented showing lncRNAs and miRNAs regulating EMT, apoptosis, cancer stem cells, DNA damage and repair. The reviewer has two minor comments:

Response: We are grateful to the Reviewer 1 for finding our study “clearly presented” and for pointing out the loopholes.

#1. Please check the spelling throughout the manuscript. For example, Figure 3, Drug resistance.

Response: We deeply apologized our mistakes in our manuscript. Revised version of the manuscript has checked carefully. Thank you for your kind notification.

#2. Lines 507-512: this paragraph does not make sense at the end of the manuscript. A paragraph to conclude the manuscript should be added.

Response: I'm sorry for the confusion. The line 507-512 is the legend of Figure 3. Now, we have changed our legend of figure 3 as follows.

“Cancer cells use a variety of mechanisms to reduce the efficacy of anticancer drugs, such as increasing transporter, inducing EMT, emitting anticancer drugs in EVs, and DNA repairing.”

Reviewer 2 Report

This manuscript is an interesting and well written synthesis about the role of ncRNAs in drug resistance in oral cancer. The main comment is that a section about ncRNAs as biomarkers of drug resistance is missing and should be added, and the different modes of action of circRNAs should be more detailed. Several errors or problems are present in the figures.

Specific comments:

Figure 1: the text in the figure is illegible

Line 70: I do not understand the mention of MALAT-1 at this place. Please clarify the sentence.

Line 124: “miR-200 family members directly bind to ZEB1 and ZEB2”. Do you mean ZEB1 and ZEB2 mRNAs? It should be specified.

Lines 140-142: “Lu MY et al. showed that the lncRNA HOTAIR suppressed the cancer stemness and metastasis of oral carcinoma stem cells through modulation of EMT. Silence of HOTAIR in oral carcinomas stem cells significantly inhibited their cancer stemness, 142 invasiveness and tumorigenicity in xenograft mouse models.”

These two sentences seem to be in contradiction. It is difficult to understand whether HOTAIR or HOTAIR silencing inhibits cancer progression. Please clarify.

Lines 149 to 157: it is boring to have 8 lines of lncRNA listing in the text. Just mention the number of lncRNAs and cite Table 1.

Line 180: “Inhibitors of apoptosis (IAPs) are known to have a baculoviral IAP…” Could you detail what is a baculoviral IAP domain? Is it possible that human cells have a baculoviral sequence if baculo is an insect virus?

Line 387: you should mention other functions of circRNAs: they are also sponges for RBPs and several of them are translated. By example FOXO3 (mentioned line 395) forms a ternary complex with cell division protein kinase 2 (CDK2) and its inhibitor p21, which blocks the CDK2 function. MALAT1 binds to both ribosome and PAX5 mRNA and thereby directly inhibits PAX5 mRNA translation by a braking mechanism.

Problem with black boxes in Figure 2

Problem in Figure 3: “Drag” resistance!

Author Response

Point-by-point responses to each of the reviewers’ comments

We are grateful to all the reviewers for their critical comments and insightful suggestions that have helped us considerably improve our paper. As indicated in the responses that follow, we have taken all of these comments and suggestions into account in the revised version of our paper. Changes in the revised manuscript are highlighted in red color.

Reviewer 2

This manuscript is an interesting and well written synthesis about the role of ncRNAs in drug resistance in oral cancer. The main comment is that a section about ncRNAs as biomarkers of drug resistance is missing and should be added, and the different modes of action of circRNAs should be more detailed. Several errors or problems are present in the figures.

Response: We are grateful to the Reviewer 1 for finding our study “an interesting and well written” and for finding out the critical point. We added several sentences about ncRNAs as biomarkers of drug resistance (line238-243).

“In addition to these miR-483-5p and MPRL, Rin Z et. al., showed that upregulation of CILA1 promotes EMT phenotype, invasiveness, and chemo-resistance in TSCC cells, whereas the inhibition of CILA1 expression induces mesenchymal-epithelial transition (MET) and chemo-sensitivity (93). The expression level of CILA1 is correlated with poor prognosis in TSCC patients. It indicated that ncRNAs, which up-regulated in cancer patients, are also useful as biomarkers.”

On the other hand, the function of circRNA is answered in #7.

Specific comments:

#1. Figure 1: the text in the figure is illegible

Response: I’m sorry to bother you. We added figure 1-3 at PDF version in this response letter. Please check it.

#2. Line 70: I do not understand the mention of MALAT-1 at this place. Please clarify the sentence.

Response: Sorry. The sentence was misleading. We have corrected it as shown below. Hope this sentence makes sense.

“Although MALAT1, one of the well-characterized lncRNAs exists, most lncRNAs have a low abundance and instability and lack typical signatures of selective constraints (20-21).”

#3. Line 124: “miR-200 family members directly bind to ZEB1 and ZEB2”. Do you mean ZEB1 and ZEB2 mRNAs? It should be specified.

Response: Thank you for your comments. We have replaced ZEB1 and ZEB2 to ZEB1 and ZEB2 mRNAs.

#4. Lines 140-142: “Lu MY et al. showed that the lncRNA HOTAIR suppressed the cancer stemness and metastasis of oral carcinoma stem cells through modulation of EMT. Silence of HOTAIR in oral carcinomas stem cells significantly inhibited their cancer stemness, 142 invasiveness and tumorigenicity in xenograft mouse models.”

These two sentences seem to be in contradiction. It is difficult to understand whether HOTAIR or HOTAIR silencing inhibits cancer progression. Please clarify.

Response: We apologized for our mistake. The lncRNA HOTAIR “promoted” the cancer stemness and metastasis of oral carcinoma stem cells through modulation of EMT. We have corrected our manuscript carefully.

#5. Lines 149 to 157: it is boring to have 8 lines of lncRNA listing in the text. Just mention the number of lncRNAs and cite Table 1.

Response: Thank you for your suggestion. We changed these sentences simply. A number of lncRNA have been identified as EMT modulators in oral cancer (48-109).

#6. Line 180: “Inhibitors of apoptosis (IAPs) are known to have a baculoviral IAP…” Could you detail what is a baculoviral IAP domain? Is it possible that human cells have a baculoviral sequence if baculo is an insect virus?

Response: Baculoviral IAP repeat (BIR) is a domain name. All members of the IAP family contain at least one BIR motif and many contain three. You can find it in a review shown here.

https://genomebiology.biomedcentral.com/articles/10.1186/gb-2001-2-7-reviews3009

#7. Line 387: you should mention other functions of circRNAs: they are also sponges for RBPs and several of them are translated. By example FOXO3 (mentioned line 395) forms a ternary complex with cell division protein kinase 2 (CDK2) and its inhibitor p21, which blocks the CDK2 function. MALAT1 binds to both ribosome and PAX5 mRNA and thereby directly inhibits PAX5 mRNA translation by a braking mechanism.

Response: As for circRNA function, we revised the sentence as shown below.

“Although several functions of circRNAs such as protein translation and binding with RNA-binding proteins are known, one of their mechanisms is functioned as a sponge of miRNAs (168-170)”

#8. Problem with black boxes in Figure 2

Response: Thank you for your comments. Same as #1, with additional figures in PDF. Hope you can see it properly.

#9. Problem in Figure 3: “Drag” resistance!

Response: We deeply apologized our mistakes in our manuscript. Revised version of the manuscript was checked carefully. Thank you for your kind notification.